# Field evaluation of capillary blood and oral-fluid HIV self-tests in the Democratic Republic of the Congo

Serge Tonen-Wolyec[1,2,3]*, Angèle Sarassoro[4], Jérémie Muwonga Masidi[5,6,7], Elie Twite Banza[7], Gaëtan Nsiku Dikumbwa[7], Dieu Merci Maseke Matondo[7], Apolinaire Kilundu[7], Luc Kamanga Lukusa[7], Salomon Batina-Agasa[3], Laurent Bélec[8,9]

1 Ecole Doctorale Régionale D'Afrique Centrale en Infectiologie Tropicale, Franceville, Gabon, 2 Department of Internal Medicine, Faculty de Medicine, University of Bunia, Bunia, The Democratic Republic of the Congo, 3 Department of Internal Medicine, Faculty of Medicine and Pharmacy, University of Kisangani, Kisangani, The Democratic Republic of the Congo, 4 Ministry of Health, Abidjan, Cote d'Ivoire, 5 Department of Clinical Biology, Faculty of Medicine, University of Kinshasa, Kinshasa, The Democratic Republic of the Congo, 6 National AIDS and STI Reference Laboratory, Kinshasa, The Democratic Republic of the Congo, 7 National AIDS and STI Control Program, Kinshasa, The Democratic Republic of the Congo, 8 Laboratory of Virology, Hôpital Européen Georges Pompidou, Paris, France, 9 University de Paris Descartes, Paris Sorbonne Cité, Paris, France

* wolyec@gmail.com

**Data Availability Statement:** All relevant data are within the manuscript and its Supporting Information files.

## Abstract

### Background

HIV self-testing (HIVST) is an additional approach to increasing uptake of HIV testing services. The practicability and accuracy of and the preference for the capillary blood self-test (Exacto Test HIV) *versus* the oral fluid self-test (OraQuick HIV self-test) were compared among untrained individuals in the Democratic Republic of the Congo (DRC).

### Methods

This multicenter cross-sectional study (2019) used face-to-face, tablet-based, structured questionnaires in a facility-based HIVST approach. Volunteers from the general public who were at high risk of HIV infection, who were between 18 and 49 years of age, and who had signed an informed consent form were eligible for the study. The successful performance and correct interpretation of the self-test results were the main outcomes of the practicability evaluation. The successful performance of the HIV self-test was conditioned by the presence of the control band. The sensitivity and specificity of the participant-interpreted results compared to the laboratory results were estimated for accuracy. Preference for either type of self-test was assessed. Logistic regression models were used to examine factors associated with participants' preference.

### Results

A total of 528 participants were included in this survey. The rate of successful performance of the HIV self-tests was high, with the blood test (99.6%) and the oral-fluid test (99.4%) yielding an absolute difference of 0.2% (95% CI: -1.8 to 1.1; *P* = 0.568). The rate of correct

**Funding:** This work was supported by the Global Fund to Fight AIDS, Tuberculosis, and Malaria and the Joint United Nations Program on HIV/AIDS. These funders played a role in data collection and analysis, but had no role in the study design, decision to publish, or preparation of the manuscript. Biosynex and OraSure Technologies, Inc., provided material support for this study in the form of HIV self-test kits and A3 printed instruction for use in the French, Lingala, and Swahili languages, but had no role in study design, data collection and analysis, decision to publish, or preparation of the manuscript.

**Competing interests:** The authors have read the journal's policy and have the following potential competing interests: Biosynex and OraSure Technologies, Inc., provided material support for this study. This does not alter our adherence to PLOS ONE policies on sharing data and materials.

interpretation of the HIV self-test results was 84.4% with the blood test *versus* 83.8% with the oral-fluid test (difference = 0.6; 95% CI: -0.2 to 1.7; *P* = 0.425). Misinterpretation (25.4% for the blood test and 25.6% for the oral-fluid test) and inability to interpret (20.4% for the blood test and 21.1% for the oral-fluid test) test results were significantly more prevalent with invalid tests. The Exacto Test HIV self-test and the OraQuick HIV self-test showed 100% and 99.2% sensitivity, and 98.9% and 98.1% specificity, respectively. Preference for oral-fluid-based HIVST was greater than that for blood-based HIVST (85.6% *versus* 78.6%; *P* = 0.008). Preference for the blood test was greater among participants with a university education (86.1%; aOR = 2.4 [95% CI: 1.1 to 4.9]; *P* = 0.016), a higher risk of HIV infection (88.1%; aOR = 2.3 [95% CI: 1.0 to 5.3]; *P* = 0.047), and knowledge about the existence of HIVST (89.3%; aOR = 2.2 [95% CI: 1.0 to 5.0]; *P* = 0.05).

## Conclusion

Our field observations demonstrate that blood-based and oral-fluid-based HIVST are both practicable approaches with a high and comparable rate of accuracy in the study setting. Although preference for the oral-fluid test was generally greater, preference for the blood test was greater among participants with a university education, a high risk of HIV infection, and knowledge about the existence of HIVST. Both approaches seem complementary in the sense that users can choose the type of self-test that best suits them for a similar result. Taken together, our observations support the use of the two HIV self-test kits in the DRC.

## Introduction

The Democratic Republic of the Congo (DRC), the largest country in Central Africa, has a relatively low HIV prevalence (1.2%) with a generalized HIV/AIDS epidemic [1, 2]. Despite progress in the scaling up of HIV testing in the country in the last 10 years, 46% of people living with HIV in the DRC remain unaware of their HIV infection, demonstrating that current efforts to meet the first "95" of the "95-95-95" UNAIDS targets remain insufficient [1, 3].

Evidence has shown that HIV testing is lacking among men and adolescents, as well as among key populations, such as female sex workers and their clients, homosexuals, transgender individuals, injection drug users, and prisoners, due to stigma, discrimination, and lack of confidentiality [4]. Thus, HIV self-testing (HIVST) is one innovation that has the potential to increase uptake of HIV testing because it offers a discreet, practical, and empowering approach [5].

According to the World Health Organization (WHO) [4], HIVST refers to a process in which a person performs an HIV test on his or her own oral fluid or blood and then interprets the result, often in a private setting, either alone or with someone he or she trusts. However, the individuals' ability to use HIV self-tests and to interpret the results correctly remains under debate [6]. Several studies in Sub-Saharan Africa showed that difficulties in correctly interpreting the self-test results represent the main barriers to effective HIVST performance [6]. Although difficulties in the collection and transfer of specimens are observed more often with blood-based self-tests than oral-fluid-based self-tests [7], the accuracy of blood-based self-tests is higher than that of oral-fluid-based self-tests due to the lower quantity of HIV antibodies in oral fluid compared with whole blood [5–9]. Furthermore, in Central Africa, the possibility exists of false-positive HIV serology results related to unspecific cross-reactivity likely

due to endemic conditions associated with uncomplicated malaria, Epstein-Barr virus infection, and other infectious diseases causing autoimmunity [10]. Furthermore, the broad genetic diversity of HIV in Central Africa, including non-B subtypes, group O, and several circulating recombinant forms, may be associated with false-negative HIV test results [11].

Although there are formal regulatory authorities, such as the WHO [12], the United States Food and Drug Administration (FDA) [13], UNITAID [14], and the Expert Review Panel for Diagnostics (ERPD) [14], tasked with approving the marketing of HIV testing devices, growing guidelines, including those of the DRC [15], suggest that only self-tests that are evaluated locally by country should be used for the program. Indeed, the verification of HIV self-test kits must take into account the local socio-cultural context, the local biodiversity of circulating HIV strains, and the relative ease of use of the test by untrained users.

The DRC has recently integrated HIVST into its HIV activity package by developing an operational manual. Previous pilot studies have shown a satisfactory rate of practicability and performance of blood-based HIVST in the DRC among the general population [9], among female sex workers [16], and among adolescents in the cities of Bunia and Kisangani [17]. However, there is no empirical evidence on the field performances of the oral-fluid HIV self-test compared to the capillary blood HIV self-test in the DRC, which would be needed to support the implementation of HIVST. Accordingly, we aimed to compare, in the field, the practicability and the accuracy of and the preference for the two self-tests using different types of specimens (whole blood and oral fluid) in the DRC.

## Material and methods

### Institutional context and ethics statement

The study was conducted as part of HIVST program implementation in the DRC and corresponds to the country's verification of HIV self-test kits. The study was conducted under the auspices of the Ministry of Public Health and in coordination with the National AIDS and STI Control Program as well as with the technical and financial support of the Global Fund to Fight AIDS, Tuberculosis, and Malaria, the Joint United Nations Program on HIV/AIDS, and country offices of the WHO. The Health Laboratories Office of the Ministry of Public Health of the DRC carried out the selection of HIV self-test kits after tendering an invitation to manufacturers wishing to register their self-tests in the DRC.

Ethical approval for this study was obtained from the Ethics Committee of the School of Public Health of the University of Kinshasa. Written informed consent was obtained from all volunteers prior to the commencement of the study. No personal information from the participants was registered to ensure anonymity. Volunteers were also informed that they could withdraw at any time from the study without any consequences.

### Study design

This was a cross-sectional study comparing the practicability and accuracy of and the preference for the blood-based HIV self-test *versus* the oral-fluid-based HIV self-test in untrained participants at risk of HIV infection using face-to-face, tablet-based, structured questionnaires in facility-based HIVST. The Exacto Test HIV self-test (Biosynex, Strasbourg, France) which uses finger-stick capillary whole blood, and the OraQuick HIV self-test (OraSure Technologies, Inc., Bethlehem, PA) which uses oral fluid to generate visually readable, qualitative, in vitro lateral flow immunoassays for the detection of antibodies to HIV-1 and HIV-2, were assessed [14]. Instructions were provided in the French, Lingala, and Swahili languages together with explanatory pictures.

## Study setting

This multicentric survey was carried out in December 2019 in the city of Kinshasa, the capital of the DRC, located in the west of the country, and Kindu, the capital city of the province of Maniema, located in the east of the country. The choice of these cities was justified by their easier accessibility, high prevalence of HIV, and different socio-cultural and geographical contexts, as recommended by the WHO [18–20]. A total of 14 study sites, integrating HIV testing and care settings, were selected for the study, including eight sites in Kinshasa (*Marechal*, *Bomoi*, *Elonga*, *Kimia*, *Matonge*, and *Saint Joseph* Health Centers; *Kalembelembe* and *Kimbondo* Pediatric Hospitals) and six sites in Kindu (*Lumbulumbu*, *Kasuku-2*, *Sokolo*, and *Mikonde* Health Centers; *Kindu* and *Alunguli* General Referral Hospitals).

## Study population and recruitment

All participants were volunteers who were successively recruited from the general public at the study sites. The study included community members who were visiting health care facilities for voluntary counseling and testing (VCT) or provider-initiated HIV counseling and testing (PICT), or prevention of mother-to-child transmission of HIV (PMTCT) clinics during the study period. Eligible participants were between 18 and 49 years of age, were at high risk of HIV infection, were unaware of their HIV status, and were able to give written informed consent. Individuals on antiretroviral treatment or pre-exposure prophylaxis, or those who did not meet the study criteria, were excluded. High risk for HIV infection was defined as a history of unprotected sex with one or more partners in the past six weeks as well as exposure to any of the following high-risk factors in the previous six months: multiple (*i.e.* ≥2) partners, homosexual intercourse (asked of men), receipt of gifts, cash, or other compensation in exchange for sex (asked of women), or infection with another sexually transmitted disease. Individuals exposed to any of these factors were classified as "high risk"; the remaining participants were classified as "low risk" if they did not report any sexual activity in the past six weeks, and as "moderate risk" otherwise [21, 22].

Based on the guideline for the evaluation of HIV testing technologies in Africa [23], the minimum sample size needed to provide 95% confidence intervals of less than ± 2% for an estimated sensitivity of 99% and a specificity of 98% was assessed to be approximately 200 HIV-positive and 200 HIV-negative specimens.

## Data collection tool

A face-to-face, tablet-based, structured questionnaire was used to collect information on socio-demographics, self-reported sexual behavior, HIV testing history, knowledge of available HIVST approaches, and HIV risk perceptions. All data on the observation of manipulation, interpretation of the results, satisfaction, preference, and field laboratory results were recorded on the tablet. The questionnaire forms for each stage of the survey were previously parametrized on tablets using the Open Data Kit Collect (ODK-Collect, Googleplex, Mountain View, USA). Data from reference laboratory analyses were recorded on reference sheets.

The main study outcomes were the practicability and accuracy of and the preference for blood-based HIV self-test *versus* oral-fluid-based HIV self-tests among participants.

**Practicability.** Practicability was evaluated by assessing participants' ability to use the finger-stick whole-blood and oral-fluid self-tests autonomously or with verbal assistance to obtain and correctly interpret valid HIV self-test results. Obtaining a valid result, which would signify the successful performance of the HIV self-tests, was conditioned by the presence of the control band. The satisfaction criteria were assessed using a quantitative Likert scale ranging from 1 (very difficult) to 4 (very easy) [24]. The satisfaction questionnaire, which concerned experiences with the self-tests, allowed for the collection of information regarding the understanding

of the instruction for use, the identification of the different components of the HIV self-test kits, the sample collection and transfer, the use of the HIV self-test, the interpretation of HIV self-test results, and the ability to overcome the difficulties encountered.

**Accuracy.** Based on the WHO requirements for pre-qualification [19, 25], the field sensitivity and specificity of the results of the HIV self-tests performed and interpreted by the participants compared to the baseline HIV test results (laboratory results) were estimated to evaluate the accuracy of the self-tests in the hands of untrained participants. The baseline HIV test results were derived from a three-test algorithm. The positive predictive values (PPVs) and negative predictive values (NPVs) were calculated in consideration of the prevalence of HIV in this series.

**Preference.** Hypothetical preference for different types of self-tests were sought as an exit questionnaire.

## Field procedures

After receiving pre-test counseling, eligible individuals were screened by the Determine HIV-1/2 (Alere Medical Co. Ltd., Matsudo-shi, Chiba-ken, Japan) test, a triage test and an initial HIV test according to the national HIV screening algorithm. Thus, all persons diagnosed initially as HIV positive with the Determine HIV-1/2 (Alere Medical Co. Ltd) and a preliminary control group (1:1 ratio) randomly selected from persons who were HIV negative were included. After receiving a brief explanation of the objectives and purpose of the study, the roles of the observer, and the possibility of receiving verbal assistance, the selected participants were asked to sign an informed consent form. After completion of the preliminary questionnaire and the random allocation of which self-test would to be taken first, the participants were asked to move to a private setting in which the instructions for use of the two self-tests would be given in their preferred language (French, Lingala or Swahili).

**Observation of manipulation.** In the private setting, after reading and declaring their understanding of the instructions for use of the two self-tests, the participants were asked to consecutively perform the two self-tests by themselves but in front of a trained observer, who would record on a tablet each step of the test in terms of its success, difficulty, errors, and need for verbal assistance (mimicking telephone support). The observation period began with the opening of the self-test box and ended at the migration stage. At the end of the session, the participant was asked to fill out the satisfaction questionnaire.

**Interpretation of the results.** For each self-test, after reading and interpreting the result of the performed self-test, the participants were introduced to five other standardized tests (which were designed to produce two positive, two negative, and one invalid result), which they were asked to interpret after successive random selection. All tests interpreted by the participants were also interpreted by trained observers to determine the expected results. At the end of the session, each participant was asked to fill out a satisfaction questionnaire concerning the interpretation of the self-test results and preference.

Finally, the participant moved to the next room with trained staff members for confirmatory HIV testing and blood sampling. At the study site, the participants were told to consider only the results of HIV testing performed according to the national HIV screening algorithm because the HIV self-tests were administered solely for research purposes. Thus, individuals who tested positive for HIV based on the national HIV testing algorithm were advised to seek post-test counseling and care.

## Laboratory procedures

At the five inclusion sites, after using the Determine HIV-1/2 test (Alere Medical Co. Ltd.) as the first test, the Uni-Gold HIV test (Trinity Biotech Manufacturing Ltd., Bray, Co. Wicklow,

Ireland) was used as the second, confirmatory test according to the national algorithm of the DRC Ministry of Public Health [15]. The HIV testing strategy based on two consecutive rapid tests with the Determine HIV-1/2 rapid immunochromatographic test as a screening test and the Uni-Gold HIV test as a confirmatory test was previously validated in the Central African Republic, a neighboring country [26]. The sensitivity and NPVs of this HIV testing strategy were both 100%, while their specificity and PPVs were similar (>98%) [26].

For the final HIV baseline results, blood was drawn by venipuncture from each participant on site and placed into EDTA tubes, after which it was centrifuged at 1,000 rpm for 15 min; the plasma was then aliquoted and kept frozen at -4°C until use at the National AIDS and STI Reference Laboratory (*Laboratoire National de Référence du SIDA et STI; "LNRS"*) of Kinshasa, DRC. All plasma samples were re-tested in the *LNRS* with the Determine HIV-1/2 (Alere Medical Co. Ltd.) and the Uni-Gold HIV (Trinity Biotech Manufacturing Ltd.) HIV tests. The results were determined to be HIV negative or HIV positive when indicated as such by both HIV tests. All discordant plasmas were further tested with the INNO-Lia HIV I/II Score (Fujirebio Europe NV, Ghent, Belgium). Plasma was considered to be positive when a band was readable for at least three markers—p24, p17, p31, gp41, gp36, sgp120, sgp 105 –and negative otherwise.

### Data management and analysis

All collected data were stored in an Excel database and analyzed using IBM SPSS Version 20 (Chicago, IL). Quantitative data were analyzed with descriptive statistics using the mean (standard deviation) or median (interquartile range) for a normal or skewed distribution, respectively, after which the proportions of all categorical variables were calculated for the qualitative data. The usability index was calculated as the arithmetic mean of all the correct answers from when the manipulation was observed. Youden's index (J) was calculated via the following formula: $J = sensitivity + specificity - 1$. The PPVs and NPVs of the self-tests were calculated by the Bayes' formula ($PPV = SePr/[SePr + (1-Sp)(1-Pr)]$; $NPV = Sp(1-Pr)/[Sp(1-Pr) + (1-Se)(Pr)]$) taking into account the prevalence (Pr) of HIV in our series. The Wilson score interval was used to estimate the 95% confidence intervals (CI). Mac Nemar's chi-squared pairing test was used to compare the results related to the practicability and accuracy of and the preference for the Exacto Test HIV self-test *versus* the OraQuick HIV self-test. The means and standard deviations for the Likert-scale data were compared between the two self-tests using the paired Wilcoxon test. Cohen's κ coefficients estimated the concordance between the results read by participants in connection with the expected results read by trained observers. Finally, when identifying the independent predictors of the participants' preference for the Exacto Test HIV self-test and/or the OraQuick HIV self-test, variables with a *P*-value < 0.2 in bivariate analysis and variables identified from the literature were entered into a full multivariate logistic regression model. Adjusted odds ratios (aOR) and their 95% CIs measured the strength of the statistical associations. A *P*-value < 0.05 was considered as statistically significant.

### Results

### Study participants

A total of 9,776 people at high risk of HIV infection were assessed for eligibility, 127 of whom were excluded (Fig 1). After triage evaluation, 528 participants were ultimately included in the study. The baseline characteristics of the study participants are depicted in Table 1. In brief, female participants (56.6%), participants between 18 and 29 years of age (42.8%), participants who were single (61.9%), participants who were unemployed (54%), and participants who were attending college or technical school (58%) were the most representative. The majority of

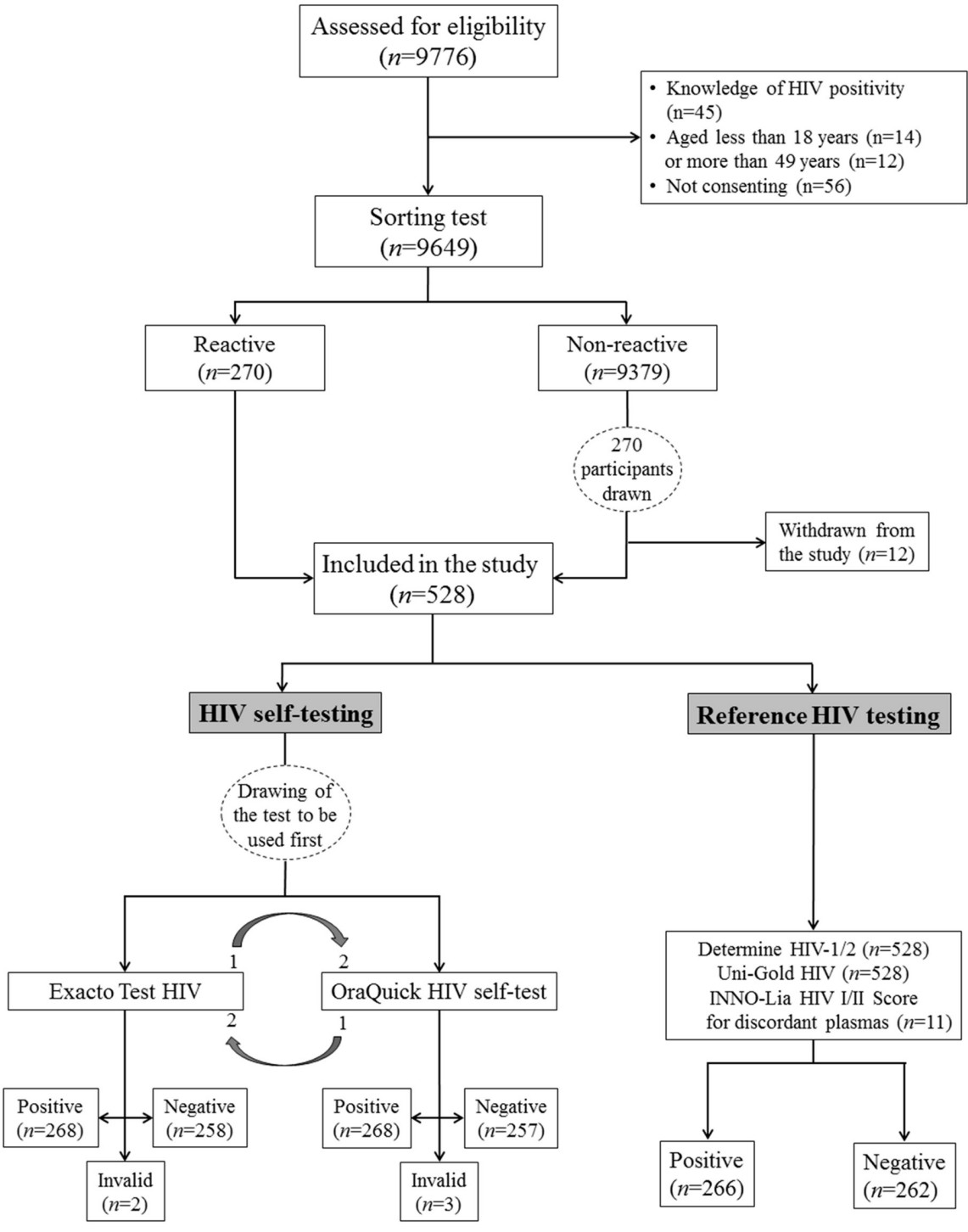

**Fig 1. Flow chart showing the recruitment of the study participants, results of HIV testing by HIV self-testing using the Exacto Test HIV self-test and the OraQuick HIV self-test, and baseline HIV test results (laboratory results).**

**Table 1. Characteristics of study participants.**

| Characteristic | Positive sorting test | Negative sorting test | Overall |
|---|---|---|---|
| | N = 258 [*number (%)*] | N = 270 [*number (%)*] | N = 528 [*number (%)*] |
| *Age (years)* | | | |
| 18–29 | 137 (53.1) | 89 (33.0) | 226 (42.8) |
| 30–39 | 69 (26.7) | 82 (30.4) | 151 (28.6) |
| 40–49 | 52 (20.7) | 99 (36.7) | 151 (28.6) |
| Means (SD) | 30.2 (8.7) | 34.7 (9.2) | 32.5 (9.2) |
| *Gender* | | | |
| Male | 118 (45.7) | 108 (40.0) | 226 (42.8) |
| Female[#] | 139 (53.9) | 160 (59.3) | 299 (56.6) |
| Transgender | 1 (0.4) | 2 (0.7) | 3 (0.6) |
| *Partnership and civil status* | | | |
| Single | 168 (65.1) | 159 (58.9) | 327 (61.9) |
| Married/partnered | 90 (34.9) | 111 (41.1) | 201 (38.1) |
| *Occupation* | | | |
| Student | 67 (26.0) | 21 (7.8) | 88 (16.7) |
| Employed | 74 (28.7) | 81 (30.0) | 155 (29.3) |
| Unemployed | 117 (45.3) | 168 (62.2) | 285 (54.0) |
| *Residence* | | | |
| Kinshasa | 155 (60.1) | 158 (58.5) | 313 (59.3) |
| Kindu | 103 (39.9) | 112 (41.5) | 215 (40.7) |
| *Educational level* | | | |
| No formal education/ completing primary school | 43 (16.7) | 71 (26.3) | 114 (21.6) |
| Attending college or technical school | 139 (53.9) | 167 (61.9) | 306 (58.0) |
| University | 76 (29.5) | 32 (11.9) | 108 (20.5) |
| *Religion* | | | |
| Catholic | 125 (48.4) | 103 (38.1) | 228 (43.2) |
| Protestant | 62 (24.0) | 72 (26.7) | 134 (25.4) |
| Islam | 22 (8.5) | 25 (9.3) | 47 (8.9) |
| Others | 49 (19.0) | 70 (25.9) | 119 (22.5) |
| *Risk of HIV infection[£]* | | | |
| Low risk | 191 (74.0) | 204 (75.6) | 395 (74.8) |
| Moderate risk | 28 (10.9) | 38 (14.1) | 66 (12.5) |
| High risk | 39 (15.1) | 28 (10.4) | 67 (12.7) |
| *Previously tested for HIV* | | | |
| Never tested | 149 (57.8) | 148 (54.8) | 297 (56.2) |
| Ever tested | 109 (42.2) | 122 (45.2) | 231 (43.8) |
| *knowledge about the existence of HIVST* | | | |
| No | 181 (70.2) | 188 (69.6) | 369 (69.9) |
| Yes | 77 (29.8) | 82 (30.4) | 159 (30.1) |
| *Previously self-tested for HIV* | | | |
| Never self-tested | 222 (86.0) | 232 (85.9) | 454 (86.0) |
| Ever self-tested | 36 (14.0) | 38 (14.1) | 74 (14.0) |

[#] Among women, 28 (5.3%) were pregnant;

[£] High risk for HIV infection was defined as a history of unprotected sex with one or more partners in the past six weeks as well as exposure to any of the following high-risk factors in the previous six months: multiple (i.e. ≥2) partners, homosexual intercourse (asked of men), receipt of gifts, cash, or other compensation in exchange for sex (asked of women), or infection with another sexually transmitted disease. Individuals exposed to any of these factors were classified as "high risk"; the remaining participants were classified as "low risk" if they did not report any sexual activity in the past six weeks, and as "moderate risk" otherwise.

participants had never been tested for HIV (56.2%) and had no knowledge of the existence of HIVST (69.9%).

## Practicability

The majority (52.8%) of participants used the instructions for use in the French language, while 29.2% and 18.0% used the instructions for use in the Lingala or Swahili language, respectively.

**Observation of manipulation.** Analytical results of observation of manipulation for each self-test are depicted separately in detail in Tables 2 and 3. Overall, most of the self-test steps were adhered to and well-executed by the participants. However, the use of the lancing device for blood-based HIVST and gingival specimen collection for oral-fluid-based HIVST required the most frequent verbal assistance in 28.4% and 20.3% of cases, respectively (Table 2). Thus, success in obtaining a valid test result was observed in 99.6% and 99.4% (difference = +0.2%; 95% CI: −1.8 to +1.1; $P = 0.568$) of cases when participants performed the Exacto Test HIV self-test and the OraQuick HIV self-test with an overall usability index of 91.2% and 95.1% (difference = -3.9; 95% CI: −5.9 to +0.1; $P = 0.093$), respectively (Table 3). Only two (0.4%) participants using the blood test and three (0.6%) participants using the oral-fluid test failed to obtain an interpretable result with a readable control band.

Although there were no statistical differences in the practicability of the two tests (Table 3), satisfaction regarding several aspects of the observation of manipulation ("identification of the different components of the HIV self-test kits": 3.30 *versus* 3.37, $P < 0.001$; "sample collection": 3.19 *versus* 3.39, $P < 0.001$; "overall use of the HIV self-test": 3.32 *versus* 3.52, $P < 0.001$; "ability to surmount the difficulties encountered": 3.29 *versus* 3.42, $P < 0.001$) was statistically higher for the oral-fluid test than for the blood test (Table 4). Finally, 23.1% of participants declared that they did not trust the results of the oral-fluid test, whereas 25.0% said they were afraid when using the lancet of the capillary blood self-test.

**Interpretation of the results.** A total of 6,336 tests (3,168 for each self-test) were read and interpreted by the 528 participants; 5,117 (80.8%; 95% CI: 79.8 to 81.8) tests were correctly interpreted, whereas 967 (15.2%; 95% CI: 14.4 to 16.2) were misinterpreted. For 252 (4.0%; 95% CI: 3.5 to 4.5) tests, participants were unable to read and interpret the test results. Cohen's κ coefficients between the results of reading by participants and the expected reference results were 0.74 and 0.71 for the blood test and oral-fluid test, respectively. The rates of correct interpretation of the self-test results were similar between the blood test and the oral-fluid test (84.4% *versus* 83.8%; difference = 0.6; 95% CI: -0.2 to 1.7; $P = 0.425$) (Table 3). Misinterpretation (25.4% for the blood test and 25.6% for the oral-fluid test) and inability to interpret (20.4% for the blood test and 21.1% for the oral-fluid test) test results were significantly more prevalent with invalid tests. The detailed results regarding the misinterpretations of the tests are depicted in Fig 2.

When extracting the results of the tests performed and interpreted by the participants, 268 (50.7%), 258 (48.9%), and two (0.4%) participants had, respectively, interpreted their results as positive, negative, or invalid with the blood self-test; whereas 268 (50.7%), 257 (48.7%), and three (0.6%) participants had interpreted their results as positive, negative, or invalid with the oral fluid self-test, respectively. Two participants misinterpreted positive results (with a low-intensity test band) from the Exacto Test HIV self-test as negative; for the OraQuick HIV self-test, three participants misinterpreted positive results as negative, and two participants misinterpreted negative results as positive. All invalid results (n = 5) of the HIV self-tests in this series were correctly interpreted. Thus, the Cohen's κ coefficients between the results of

**Table 2. Analytical results of the manipulation observation concerning the ability of the 528 study participants to correctly use each step of the HIV self-test autonomously or with oral assistance.**

| Usability checklist* | Successful manipulation | Need for verbal assistance |
|---|---|---|
| | Yes [*number (%)*] | Yes [*number (%)*] |
| **Exacto Test HIV self-test** | | |
| 1. Easy identifying the different components of the kit? | 484 (91.7) | 44 (8.3) |
| 2. Washing the hands? | 491 (93.0) | - |
| 3. Removing correctly the test cassette from the bag? | 510 (96.6) | - |
| 4. Opening correctly the diluent vial? | 517 (97.9) | - |
| 5. Disinfecting properly the finger? | 450 (85.2) | - |
| 6. Wiping correctly residual alcohol with the compression swab? | 430 (81.4) | - |
| 7. Lancing the finger without difficulty? | 378 (71.6) | 150 (28.4) |
| 8. Forming a blood droplet without difficulty? | 501 (94.9) | - |
| 9. Proper placing in contact the drop of blood with the sampler tip? | 446 (84.5) | 82 (15.5) |
| 10. Checking that the sampler tip was filled with blood? | 480 (90.9) | - |
| 11. Transferring the blood into the SQUARE well of the test cassette? | 527 (99.8) | 5 (0.9) |
| 12. Shedding two drops of diluent in the ROUND well of the test cassette? | 522 (98.9) | - |
| 13. Obtaining an interpretable result at the end of the process?# | 526 (99.6) | NA |
| *Usability index (%)£* | 91.2 | NA |
| **OraQuick HIV Self-test** | | |
| 1. Easy identifying the different components of the kit? | 510 (96.6) | 18 (3.4) |
| 2. Easy finding the test kit containing two packets? | 521 (98.7) | 7 (1.3) |
| 3. Removing the test tube from the pouch without difficulty? | 523 (99.1) | - |
| 4. Removing the cap from the test tube without difficulty? | 496 (93.9) | 18 (3.4) |
| 5. Following instructions not to spill the liquid or drink it? | 528 (100) | - |
| 6. Sliding the test tube into the stand without difficulty? | 482 (91.3) | 46 (8.7) |
| 7. Removing the test device from the pouch without touching the flat pad? | 512 (97.0) | - |
| 8. Collecting correctly the sample? | 421 (79.7) | 107 (20.3) |
| 9. Placing correctly the test device in the test tube correctly? | 468 (88.6) | 60 (11.4) |
| Obtaining an interpretable result at the end of the process?# | 525 (99.4) | NA |
| *Usability index (%)£* | 95.1 | NA |

* The majority (52.8%) of participants used the instructions for use in the French language, while 29.2% and 18% used the instructions for use in the Lingala and Swahili language, respectively;

# The result was considered interpretable when a control band was readable after the migration time recommended by the manufacturers;

£ The usability index was calculated as the arithmetic mean of all the correct answers from when the manipulation was observed.

NA: Not Applicable

reading by participants *versus* observers were 0.99 and 0.98 for the blood test and oral-fluid test, respectively.

## Accuracy

The results of the accuracy of participant-interpreted HIV self-test results are depicted in Fig 1 and Table 3.

**Table 3. Comparison of the practicability and accuracy of and preference for the Exacto Test HIV self-test *versus* the OraQuick HIV self-test in the hand of untrained users.**

| Characteristic | Exacto Test HIV self-test | OraQuick HIV self-test | Difference | P-value[*] |
|---|---|---|---|---|
| | % (95% CI)[#] | % (95% CI) [#] | % (95% CI) [#] | |
| **Practicability:** | | | | |
| Easy identification of the components of the test kit | 91.7 (89.0 to 93.8) | 96.6 (94.7 to 97.8) | -4.9 (-7 to 0.3) | 0.073 |
| Successful performance[β] | 99.6 (98.6 to 99.9) | 99.4 (94.4 to 97.7) | 0.2 (-1.8 to 1.1) | 0.568 |
| Usability index | 91.2 (88.5 to 93.3) | 95.1 (92.9 to 96.6) | -3.9 (-5.9 to 0.1) | 0.093 |
| Correct interpretation of the HIVST results | 84.4 (83.1 to 85.6) | 83.8 (82.4 to 85.1) | 0.6 (-0.2 to 1.7) | 0.425 |
| **Accuracy:** | | | | |
| Sensitivity | 100 (99.2 to 100) | 99.2 (98.0 to 99.7) | 0.8 (-0.3 to 2.0) | 0.721 |
| Specificity | 98.9 (97.6 to 99.5) | 98.1 (96.5 to 99.0) | 0.8 (-0.3 to 2.0) | |
| Youden index[$] | 98.9 (97.6 to 99.5) | 97.3 (95.5 to 98.4) | 1.2 (-0.6 to 3.4) | |
| PPV[£] | 71.6 (67.6 to 75.3) | 60.2 (56.0 to 64.3) | 11.4 (8.9 to 14.4) | NA |
| NPV[£] | 100 (99.2 to 100) | 99.98 (99.2to 100) | 0.02 (-0.99 to 0.1) | NA |
| **Personal preference on type of self-test[&]** | 78.6 (74.9 to 81.9) | 85.6 (82.3 to 88.3) | -7.0 (-12.1 to -1.9) | 0.008 |

[*] *P*-value calculated using Mac Nemar's test of paired data;

[#] The 95% confidence intervals were calculated using the Wilson score bounds;

[β] The successful performance was conditioned by the presence of a control band readable after the migration step;

[$] Youden's index (J) was calculated via the following formula: $J = sensitivity + specificity - 1$;

[£] The PPVs and NPVs of the self-tests were calculated by the Bayes' formula (PPV = SePr/[SePr+(1–Sp)(1–Pr)]; NPV = Sp(1–Pr)/[Sp(1–Pr)+(1–Se)(Pr)]) taking into account the prevalence (Pr) of HIV in our series estimated at 2.76% (266 confirmed positive among 9649 screened persons);

[&] Overall, 88% preferred that both types of tests be available.

CI: Confidence Interval; NA: Not Applicable; NPV: Negative Predictive Value; PPV: Positive Predictive Value

For the Exacto Test HIV self-test, no participant with a negative self-test result was found to be HIV positive in laboratory HIV testing; in contrast, three participants with a positive self-test result were found to be HIV negative in laboratory testing. Thus, based on participant interpretation of the HIV self-test results, the sensitivity, specificity, PPVs, and NPVs of the blood test were estimated to be 100%, 98.9%, 71.6%, and 100%, respectively.

**Table 4. Results of the satisfaction questionnaire.**

| Satisfaction | Exacto Test HIV self-test | OraQuick HIV self-test | P-value[£] |
|---|---|---|---|
| | [*mean (SD)*][*] | [*mean (SD)*][*] | |
| 1. Overall understanding of the instruction for use | 3.47 (0.72) | 3.51 (0.67) | 0.031 |
| 2. Identification of the different components of the HIV self-test kits | 3.30 (0.83) | 3.37 (0.82) | < 0.001 |
| 3. Sample collection | 3.19 (0.83)[#] | 3.39 (0.74) | < 0.001 |
| 4. Sample transfer | 3.48 (0.63) | 3.57 (0.59) | 0.002 |
| 5. Overall use of the HIV self-test | 3.32 (0.70) | 3.52 (0.57) | < 0.001 |
| 6. Interpretation of a *"positive"* HIV self-test result | 3.62 (0.56) | 3.62 (0.55)[β] | 0.878 |
| 7. Interpretation of a *"negative"* HIV self-test result | 3.62 (0.55) | 3.63 (0.54) | 0.625 |
| 8. Interpretation of a *"invalid"* HIV self-test result | 2.97 (1.06) | 2.96 (1.06) | 0.751 |
| 9. Ability to overcome the difficulties encountered | 3.29 (0.71) | 3.42 (0.62) | < 0.001 |

[*] The satisfaction criteria were assessed using a quantitative Likert scale ranging from 1 (very difficult) to 4 (very easy);

[£] The comparison of the means was done using the paired Wilcoxon nonparametric test;

[#] 25% (*n* = 132) of participants said they were afraid when using the lancet;

[β] 23.1% (*n* = 122) of participants declared that they did not trust the results of the oral-fluid test.

SD: Standard Deviation

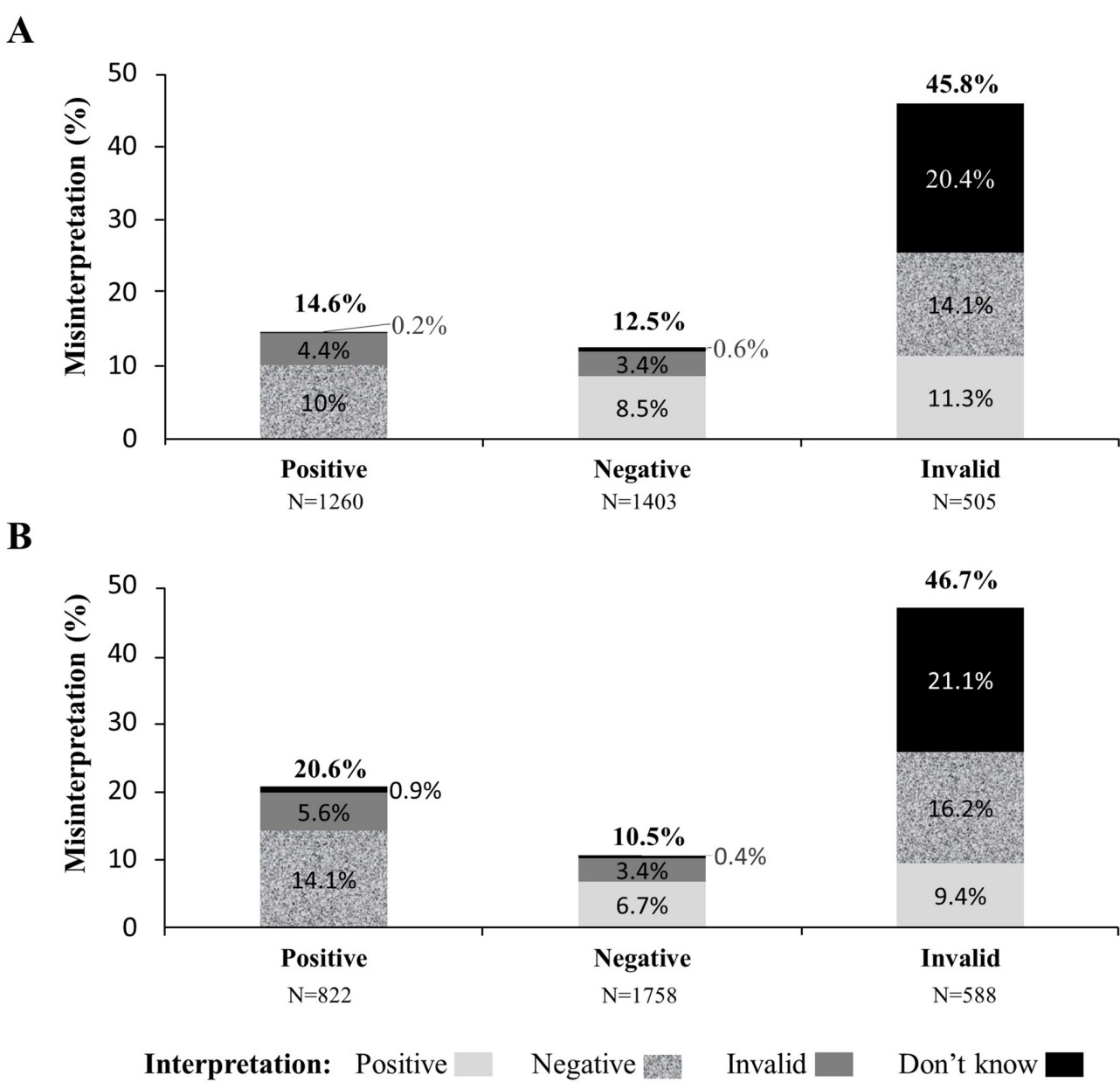

**Fig 2. Stacked columns showing the percent of the misinterpreted self-tests results by using the Exacto Test HIV self-test [A] and the OraQuick HIV self-test [B].** The heights of the columns indicate the overall percent of the misinterpreted HIV self-tests results and inability to interpret; the hatched components of the columns indicated the type of misinterpreting and inability to interpret.

For the OraQuick HIV self-test, two participants with a negative self-test result were found to be HIV positive in laboratory HIV testing, whereas five participants with a positive self-test result were found to be HIV negative in laboratory testing. Thus, based on participant interpretation of the HIV self-test results, the sensitivity, specificity, PPVs, and NPVs of the oral-fluid self-test were estimated to be 99.2%, 98.1%, 60.2%, and 99.98%, respectively.

No statistical differences were observed when comparing the distribution of values between the two self-tests (Table 3).

**Table 5. Multivariate regression analysis of factors possibly associated with the preference of the Exacto Test HIV self-test and the OraQuick HIV self-test among the 528 study participants.**

| Characteristic | Preference for Exacto Test HIV self-test | | | Preference for OraQuick HIV self-test | | |
|---|---|---|---|---|---|---|
| | Yes % | aOR (95% CI) | P-value* | Yes % | aOR (95% CI) | P-value* |
| *Educational level* | | | | | | |
| No formal education/ completing primary school | 71.1 | 1 | 1 | 80.7 | 1 | 1 |
| Attending college or technical school | 78.8 | 1.7 (0.9 to 4.2) | 0.107 | 86.9 | 1.2 (0.6 to 2.3) | 0.598 |
| University | 86.1 | 2.4 (1.1 to 4.9) | 0.016 | 87.0 | 1.2 (0.6 to 2.7) | 0.681 |
| *Risk of HIV infection*£ | | | | | | |
| Low risk | 75.4 | 1 | 1 | 91.4 | 1 | 1 |
| Moderate risk | 87.9 | 1.0 (0.3 to 3.0) | 0.543 | 68.2 | 0.2 (0.04 to 2.1) | 0.998 |
| High risk | 88.1 | 2.3 (1.0 to 5.3 | 0.047 | 68.7 | 0.2 (0.1 to 0.4) | < 0.001 |
| *Previously tested for HIV* | | | | | | |
| Never tested | 75.1 | 1 | 1 | 85.2 | 1 | 1 |
| Ever tested | 83.1 | 1.3 (0.7 to 2.3) | 0.448 | 86.1 | 1.7 (0.7 to 3.9) | 0.245 |
| *Knowledge about the existence of HIVST* | | | | | | |
| No | 74.0 | 1 | 1 | 85.9 | 1 | 1 |
| Yes | 89.3 | 2.2 (1.0 to 5.0) | 0.05 | 84.9 | 0.6 (0.2 to 1.7) | 0.371 |
| *Previously self-tested for HIV* | | | | | | |
| Never self-tested | 76.0 | 1 | 1 | 85.7 | 1 | 1 |
| Ever self-tested | 94.6 | 3.2 (0.96 to 10.4) | 0.058 | 85.6 | 0.9 (0.4 to 2.4) | 0.920 |

*P-value calculated using regression analysis;

£ High risk for HIV infection was defined as a history of unprotected sex with one or more partners in the past six weeks as well as exposure to any of the following high-risk factors in the previous six months: multiple (i.e. ≥2) partners, homosexual intercourse (asked of men), receipt of gifts, cash, or other compensation in exchange for sex (asked of women), or infection with another sexually transmitted disease. Individuals exposed to any of these factors were classified as "high risk"; the remaining participants were classified as "low risk" if they did not report any sexual activity in the past six weeks, and as "moderate risk" otherwise.

aOR: adjusted Odds ratios; CI: Confidence interval.

## Preference

Overall, the hypothetical preference for oral-fluid-based HIVST (85.6% [95% CI: 82.3 to 88.3]) was greater than that for blood-based HIVST (78.6% [95% CI: 74.9 to 81.9]), yielding an absolute difference of -7.0% (95% CI: -12.1 to -1.9; *P* = 0.008). Nevertheless, 88% of participants preferred for both types of tests to be available.

Regarding factors possibly associated with the preference of participants for these two types of HIV self-tests, multivariate analysis followed by logistic regression showed that the preference for the blood-based self-test was greater among participants with a university education (86.1%; aOR = 2.4 [95% CI: 1.1 to 4.9]; *P* = 0.016), a high risk of HIV infection (88.1%; aOR = 2.3 [95% CI: 1.0 to 5.3]; *P* = 0.047), and knowledge about the existence of HIVST (89.3%; aOR = 2.2 [95% CI: 1.0 to 5.0]; *P* = 0.05) (Table 5). However, the preference for the oral-fluid self-test was lower among participants with a high risk of HIV infection (68.7%; aOR = 0.2 [95% CI: 0.1 to 0.4]; *P* < 0.001). Although the preference for blood-based self-testing was very high (94.6%) among participants with previous HIV testing, no such association could be established in multivariate analysis (*P* = 0.058).

## Discussion

In the present study, the rate of successful performance of the HIV self-tests was high for both the blood-based self-test and the oral-fluid-based self-test. Difficulty in interpreting invalid

results was observed for both tests. Although there was no statistical difference in the manipulation and interpretation of the results of both HIV self-tests, the satisfaction of participants was higher for the oral-fluid-based self-test. The Exacto Test HIV self-test and the OraQuick HIV self-test both showed high levels of sensitivity and specificity. Overall, the preference for the oral-fluid-based self-test was slightly greater than that for the blood-based self-test (85.6% *versus* 78.6%). The preference for the blood-based self-test was greater among participants with a university education (86.1%), a high risk of HIV infection (88.1%), and knowledge about the existence of HIVST (89.3%). Finally, the majority (88%) of participants preferred for both types of self-tests to be available.

Figueroa *et al*. previously reported fewer difficulties in the performance of the oral- fluid-based self-test than in the performance of the blood-based self-test [6]. While the oral-fluid self-test has the advantage of being non-invasive, the blood-based self-test presents difficulties related to the self-collection and transfer of blood samples [27]. However, a recent study reported that the ease of and confidence in the ability to perform blood-based self-tests appears to increase with greater HIVST testing experience [7]. In our series, apart from the differences observed in terms of satisfaction and preference, the participants' ability to complete the self-tests was virtually comparable between the two tests.

In the present series, confidence in the test results of oral-fluid-based HIVST was lower in people at high risk of HIV infection. Thus, 23.1% of study participants declared that they did not trust the results of the oral-fluid self-test. Unfortunately, our protocol did not include focused research on the causes of this lack of confidence in the results of the oral-fluid self-test. Concerns over HIV self-test results were also reported in South Africa [8]. There is a need to avoid stigma before introducing an oral-fluid self-test in the DRC because some people may believe that since oral secretions are used to test for HIV, there may be a risk of oral transmission.

Due to the low educational level of many individuals in the key population, such as adolescents, in Central Africa [16, 17, 28, 29], difficulties in interpreting test results constitute a major concern for the safe implementation of HIVST. For instance, should an individual misinterpret an invalid/positive test as negative, he or she could unknowingly infect others [30]. The instructions for use of HIV self-tests could be improved by including educational resources such as videos, online support, and mobile applications to limit errors in the manipulation and interpretation of test results [4, 20].

Several authors have suggested that the main concern about HIV self-tests is not the type of test but its accuracy in detecting HIV [5, 6, 31]. Although previous studies have suggested that the accuracy of blood-based self-tests is higher than that of oral-fluid self-tests due to the lower quantity of HIV antibodies in oral fluid compared with whole blood [6], our findings show that the sensitivity and specificity of both evaluated tests were comparable and met both the criteria set forth by the WHO [20] and the DRC guideline [15] (*i.e*., sensitivity $\geq$ 99% and specificity $\geq$ 98%). These findings are in line with those of previous studies, which showed that the Exacto Test HIV self-test and the OraQuick HIV self-test demonstrated similar accuracy in comparison with conventional blood-based HIV rapid tests [9, 32]. However, precautions concerning meal and water consumption prior to performing the oral-fluid self-test must be strictly adhered to in order to obtain correct results. Furthermore, performing a saliva test can be difficult in the event of dry mouth, or even dry syndrome, or in the event of hypersalivation, with a risk of false-negative results due to the deficiency of saliva or the dilution of salivary antibodies [33]. On the other hand, the accuracy of self-tests, blood-based or oral-fluid-based, can be compromised by deficient self-interpreted results, regardless of the quality of the test itself [21, 34]. In this survey, misinterpretation of self-tests performed and read by participants was more likely to occur in the case of positive results with a low-intensity test

band. This is likely due to practical difficulties incorrectly reading self-test results among some individuals, such as those with little education or visual impairments [9, 16, 28].

As about 25% of the participants reported feeling afraid to use a lancet, the preference for the oral-fluid self-test was significantly higher than that for the blood-based self-test. This finding is in line with those reported by Ritchwood *et al.* among South African youth [8]. However, in our series, a high educational level, risk-taking behavior that could lead to HIV infection, and knowledge about the existence of HIVST were factors that increased preference for the blood-based self-test. It can therefore be argued that students in the DRC, regardless of their field of study, benefit from taking HIV-related courses. These students are not only aware of the existence of HIVST but also, in the case of HIV risk, prefer to use a more accurate test, one that would not be very different from those used in the health facilities (*i.e.*, a blood test). Previous studies have also shown that, generally speaking, people most concerned about accuracy prefer to take a blood-based self-test [8].

## Strengths and limitations

Our study, which compared two self-tests that used different type of specimens, was the first of its kind to be conducted in the DRC, and thus its novelty represents a strength. Additionally, our observations highlighted the importance of: (i) implementing locally validated HIV self-test kits in the field, and (ii) making both types of self-tests available as they are comparable and complementary. Nevertheless, the study has some limitations. First, the presence of an observer may have led to bias concerning the participants' ability to perform the tests and to interpret the results. Second, our study did not assess the internal quality control of each test. Since both self-tests use a migration control system and not an immunological control, there is a very high risk of false-negative results should the sample be improperly collected and transferred. Indeed, with the migration control system, the use of the buffer alone (without a sample) can make a control band visible after migration, and therefore the result would be read as negative. This is a major limitation of HIVST, especially in the case of faulty sample collection. While these usability biases cannot be practically demonstrated with the oral-fluid self-test, the presence of blood in the relevant well can serve as another quality control argument with the blood test [9]. Furthermore, in this study, we found that the misinterpretation of test results occurred more often when participants were asked to interpret other standardized tests results that they themselves had not performed. This may justify the high rate of misinterpretation, and yet it nonetheless constitutes a limitation. Finally, selection and volunteer bias likely occurred.

In conclusion, our field observations show that blood-based and oral-fluid based HIVST are both practicable approaches with high and comparable rates of accuracy. Although the preference for the oral-fluid self-test was general greater, preference for the blood-based self-test was higher among participants who had a university education, had a greater risk of HIV infection, and possessed knowledge of the existence of HIVST. Importantly, the value of these two self-tests can be complementary since their results can be compared. Accordingly, both self-tests (Exacto Test HIV self-test and OraQuick HIV self-test) should be made available in the DRC.

## Supporting information

**S1 Appendix. Study raw data.**
(XLS)

## Acknowledgments

The authors are grateful to the volunteers for their willingness to participate in the study. We also thank the provincial AIDS and STI control coordination and various actors in the field who facilitated this work. Thanks are also due to Raphaël Dupont and Salah Azzi. Finally, the authors thank Dr. Etienne Mpoyi (WHO Contact Person), Dr. Willy Mvita (Manager of the HIV Project at the *Cellule d'Appui et de Gestion Financière [CAGF]*), Dr. If Malaba (Health Laboratories Chief manager at the DRC Ministry of Public Health), and Ms. Bintou Touré-Fadiga (Disease Fund Manager [HIV/TB] at the Global Fund to Fight AIDS, Tuberculosis, and Malaria).

## Author Contributions

**Conceptualization:** Serge Tonen-Wolyec, Angèle Sarassoro, Jérémie Muwonga Masidi, Apolinaire Kilundu, Luc Kamanga Lukusa, Salomon Batina-Agasa, Laurent Bélec.

**Data curation:** Serge Tonen-Wolyec, Angèle Sarassoro, Elie Twite Banza, Dieu Merci Maseke Matondo.

**Formal analysis:** Serge Tonen-Wolyec, Angèle Sarassoro, Jérémie Muwonga Masidi, Gaëtan Nsiku Dikumbwa, Laurent Bélec.

**Funding acquisition:** Angèle Sarassoro, Apolinaire Kilundu, Luc Kamanga Lukusa.

**Investigation:** Serge Tonen-Wolyec, Angèle Sarassoro, Dieu Merci Maseke Matondo.

**Methodology:** Serge Tonen-Wolyec, Angèle Sarassoro, Jérémie Muwonga Masidi.

**Software:** Gaëtan Nsiku Dikumbwa.

**Supervision:** Serge Tonen-Wolyec, Angèle Sarassoro, Salomon Batina-Agasa, Laurent Bélec.

**Validation:** Serge Tonen-Wolyec, Elie Twite Banza, Gaëtan Nsiku Dikumbwa, Salomon Batina-Agasa, Laurent Bélec.

**Visualization:** Serge Tonen-Wolyec, Gaëtan Nsiku Dikumbwa, Laurent Bélec.

**Writing – original draft:** Serge Tonen-Wolyec, Salomon Batina-Agasa, Laurent Bélec.

**Writing – review & editing:** Serge Tonen-Wolyec, Salomon Batina-Agasa, Laurent Bélec.

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
