## [Decision Letter · Decision Letter 0]

4 Aug 2020

PONE-D-20-17754

Field evaluation of capillary blood and oral-fluid HIV self-tests  in the Democratic Republic of the Congo

PLOS ONE

Dear Dr. Tonen-Wolyec,

Thank you for submitting your manuscript to PLOS ONE. After careful consideration, we feel that it has merit but does not fully meet PLOS ONE’s publication criteria as it currently stands. Therefore, we invite you to submit a revised version of the manuscript that addresses the points raised during the review process.

We look forward to receiving your revised manuscript.

Kind regards,

Joel Msafiri Francis, MD, MS, PhD

Academic Editor

PLOS ONE

Journal Requirements:

2. Please specify in your ethics statement whether participant consent was written or verbal. If verbal, please also specify: 1) whether the ethics committee approved the verbal consent procedure, 2) why written consent could not be obtained, and 3) how verbal consent was recorded.

Reviewers' comments:

Reviewer's Responses to Questions

**Comments to the Author**

1. Is the manuscript technically sound, and do the data support the conclusions?

Reviewer #1: Partly

Reviewer #2: Yes

2. Has the statistical analysis been performed appropriately and rigorously? 

Reviewer #1: No

Reviewer #2: Yes

3. Have the authors made all data underlying the findings in their manuscript fully available?

Reviewer #1: Yes

Reviewer #2: Yes

4. Is the manuscript presented in an intelligible fashion and written in standard English?

Reviewer #1: Yes

Reviewer #2: Yes

5. Review Comments to the Author

Reviewer #1: Introduction

1.The authors use the word "validation" in their reports. please refer to the standard definition and provide explanation whether was was done was validation or verification.

2. Throughout the paper the author has used the work "control strip" meaning a control band which is found on a RDT strip. I suggest they use the word control band instead of strip

Methods

1 Reference testing of sample was done using two rapid tests Determine HIV-1/2 and Unigold on samples which has discordant results in the two rapid assays. The authors did not provide evidence that the two assays have been assessed previously and found to be acceptable reference tests without concordant false positives, false negative results. in absence of such study, the concordant positives and discordant should be tested on more sensitive and specific assays including 4th generation EIAs and Western Blot. Without such testing, the authors should refrain from using the terms sensitivity and specificity and consider using agreement. it becomes very difficult to believe the accuracy of the two HIV self tests in absence of adequate testing of the samples using internationally acceptable reference testing.

2. The samples which had different results in the two "reference results should be labledd as discordant rather than inderterminate status.

Results

1. 28.4% and 20.3% of cases could not collect the samples correctly. This is a big limitation which has to be included in the study limitations and thoroughly discussed

2. Misinterpretation

25.4% for the blood test and 25.6% for the oral-fluid test results were wrongly interpreted. again this is a big limitation which should be discussed and taken in consideration before introducing such tests to the community

Reviewer #2: OVERALL: An important paper that is needed to inform the scale-up of HIVST in DRC. However, the paper presents a lot of outcomes, and could benefits from editing that streamlines the main study findings and language used to describe the methods/results.

MAJOR:

• Abstract – the quality of the abstract does not reflect the quality of the paper. Please revise.

• Methods – There are somethings that are presented in the results section that were not clarified in the methods, please ensure that all outcome metrics presented in the results are described in the methods.

• There are a many outcomes presented in this paper, which is great and demonstrates a very thorough investigation of the research question. However, at times, keeping track of all these outcomes is a bit confusing, especially with the many different tables. Throughout the methods and results, could be helpful to clearly map out the outcomes being measured and ensure that the results presented map to these – for example, where does satisfaction fit in?

• Re-read the paper for grammar and language, sometimes the language used to describe things is a bit unusual and inconsistent. Try to use simple, consistent language when possible.

MINOR:

Abstract:

• (Background): Suggest “HIVST is an additionally approach…”

• (Background): Suggest taking out details on the test manufacturer in the abstract – can leave this for the paper.

• (Methods): What is your study population – what was the eligibility criteria for inclusion in the study?

• (Methods): Not clear if qualitative or quantitative data was collected – please clarify.

• (Methods): You define practicability, but not accuracy and preferences (discussed in the background). Please clarify how these other outcomes were measured.

• (Methods): Not sure what you mean by matching tests, please provide more information.

• (Results): How were participants prospectively enrolled if this was a cross-sectional study?

• (Methods/Results): How are you defining a successful performance? % of certain necessary steps completed?

• (Results): What % of the tests resulted in invalid results?

• (Results): How did you measure sensitivity and specificity?

• Check punctuation and spacing between words throughout.

• (Conclusions): Consider mentioning how there was a greater preference for oral-fluid versus blood-based HIVST.

Methods:

• (Study design): How can a cross-sectional survey have a semi-structured questionnaire? These are usually used in qualitative research. Please explain.

• (Study setting): It does not seem that the locations of the study was “arbitrary” if you have a rationale for these settings, which you appear to have.

• (Study population): How did you determine that participants were at high risk of HIV infection? How did you recruit participants into the study?

• (Data collection): Same question as above about semi-structured survey. Suggest moving this information about “high risk” to the study population section above.

• (Data collection): Is correctly interpreting results a practicability measurement or an accuracy measurement?

• (Data collection); What do you mean by reference HIV testing results? Can you be more specific here?

• (Field procedures): When you screened participants using the Determine test, did you give them their test results? Or did they just receive their test results from the self-testing kits? Additionally, do you have photos of the other standardized tests you had them interpret? (include as appendices?)

• (Data management and analysis): I would move the information about the satisfaction criteria to the data collection section. Also, for your multivariable logistic regression model, what independent variables did you include in this analysis?

Results:

• (Study participants): Do you mean no knowledge of the existence of HIVST?

• (Table 1): What is the number for the overall participants not the sum of the non-reactive and reactive sorting test participants?

• (Practicability): How did you calculate the usability index? This should be in the methods section.

• (Table 2): Can the information presented in this table be simplified? Can you just include the “Yes” column for the steps and abbreviate the text describing the steps?

• (Practicability): What were the outcomes of the tests? E.g., how many were inconclusive?

• (Table 3): Can you add a description of how you calculated the Youden index to the methods section?

• (Accuracy): For this measurement, can you clarify if this is based on participant interpretation of the HIVST, or researcher interpretation of the HIVST?

Discussion:

• What discussing the accuracy of the HIVST compared to standard testing methods, can you also discuss how self-interpreted results can decrease the accuracy of the HIVST, because participants might interpret the results differently from what the test is (accurately) showing.

6. PLOS authors have the option to publish the peer review history of their article (what does this mean?). If published, this will include your full peer review and any attached files.

Reviewer #1: **Yes: **Willy Kikoka Urassa

Reviewer #2: No

---

## [Author Response · Author response to Decision Letter 0]

4 Sep 2020

Responses to journal requirements and to Reviewers 

Journal Requirements:

Our answer: We have checked that the manuscript meets the PLOS ONE’S requirements, including file names and affiliations. 

2. Please specify in your ethics statement whether participant consent was written or verbal. If verbal, please also specify: 1) whether the ethics committee approved the verbal consent procedure, 2) why written consent could not be obtained, and 3) how verbal consent was recorded.

Our answer: We have specified that informed consent was obtained from all volunteers in writing. No personal information from the participants was registered to ensure anonymity. Volunteers were also informed that they could withdraw at any time from the study without any consequences. 

Our answer: It was a mistake, we have corrected it. 

Reviewers' comments: 

Reviewer's Responses to Questions

Comments to the Author

1. Is the manuscript technically sound, and do the data support the conclusions?

Reviewer #1: Partly

Reviewer #2: Yes

Our answer: We thank the reviewers for their nice comments on our work. However, in order to acknowledge the comments raised by Reviewer #1, we have made corrections thorough the manuscript (Introduction, Methods, Results, Discussion and Conclusions sections). We hope that our manuscript is now more technically sound.

2. Has the statistical analysis been performed appropriately and rigorously?

Reviewer #1: No

Reviewer #2: Yes 

Our answer: We have corrected the statistical analysis section for clarity, and we hope that our statistical analyses are performed appropriately and rigorously.

3. Have the authors made all data underlying the findings in their manuscript fully available?

Reviewer #1: Yes

Reviewer #2: Yes

Our answer: In order to acknowledge the comments raised by Reviewers, we have provided Excel data base as supporting information, thus any qualified researcher will be able to manage and analyze this data as needed. 

4. Is the manuscript presented in an intelligible fashion and written in standard English?

Reviewer #1: Yes

Reviewer #2: Yes

Our answer: To address the concerns of Referee #2, the English of the manuscript has been fully edited by the assistance of International Research Promotion English Language Editing Services (IRP-ELES) to fully copyedit our manuscript in order to ensure proper English wording and grammar, and an editing certificate is provided. 

5. Review Comments to the Author

Answer to reviewer #1

Introduction

1.The authors use the word "validation" in their reports. please refer to the standard definition and provide explanation whether was was done was validation or verification.

Our answer: We have changed the word validation by verification. 

2. Throughout the paper the author has used the work "control strip" meaning a control band which is found on a RDT strip. I suggest they use the word control band instead of strip

Our answer: We have changed the word “control strip” by “control band”.

Methods

1 Reference testing of sample was done using two rapid tests Determine HIV-1/2 and Unigold on samples which has discordant results in the two rapid assays. The authors did not provide evidence that the two assays have been assessed previously and found to be acceptable reference tests without concordant false positives, false negative results. in absence of such study, the concordant positives and discordant should be tested on more sensitive and specific assays including 4th generation EIAs and Western Blot. Without such testing, the authors should refrain from using the terms sensitivity and specificity and consider using agreement. it becomes very difficult to believe the accuracy of the two HIV self tests in absence of adequate testing of the samples using internationally acceptable reference testing.

Our answer: We thank the reviewer for this pertinent remark. However, it should be noted that we used the DRC's national HIV serological screening algorithm, using the Determine and Unigold tests, as the final reference test. In addition, we used a third discrimination test (INNO-Lia HIV I/II Score). This test was used in the event of a discrepancy (in the field or in the reference laboratory) between the two previous tests. Note that the HIV testing strategy based on two consecutive rapid tests with the Determine HIV-1/2® rapid immunochromatographic test as a screening test and the Uni-Gold HIV test® (Trinity Biotech®, Dublin, Ireland) as a confirmatory test was previously validated in the Central African Republic, a neighboring countries very closed of the Democratic Republic of Congo (Ménard et al., Journal of Virological Methods 205; 126: 75-80). The sensitivity and negative predictive value of this HIV testing strategy were 100%. The sensitivity and negative predictive value of this HIV testing strategy were 100%. Their specificity and positive predictive values were similar (>98%).We have added this previous validation and the reference by Ménard and colleagues in the Laboratory procedures paragraph.

Reference added: Ménard D, Maïro A, Mandeng MJ, Doyemet P, Koyazegbe Td, Rochigneux C, Talarmin A. Evaluation of rapid HIV testing strategies in under equipped laboratories in the Central African Republic. J Virol Methods. 2005 Jun;126(1-2):75-80.

2. The samples which had different results in the two "reference results should be labledd as discordant rather than inderterminate status.

Our answer: To acknowledgment reviewer’s remark, we have corrected this concern in revised version of our manuscript. 

Results

1. 28.4% and 20.3% of cases could not collect the samples correctly. This is a big limitation which has to be included in the study limitations and thoroughly discussed

Our answer: We have discussed this issue and included it in the paragraph on study limitations. 

2. Misinterpretation

25.4% for the blood test and 25.6% for the oral-fluid test results were wrongly interpreted. again this is a big limitation which should be discussed and taken in consideration before introducing such tests to the community

Our answer: We have discussed this issue. In any case, other tools such as video notice, internet, etc. are needed to reduce these limitations of the feasibility of self-testing. 

Answer to reviewer #2

OVERALL: An important paper that is needed to inform the scale-up of HIVST in DRC. However, the paper presents a lot of outcomes, and could benefits from editing that streamlines the main study findings and language used to describe the methods/results.

Our answer: We thank the reviewer for bringing positive reviews to our paper. We have modified all concerns regarding the methods and results. We feel that our methods are explicit in explaining our results.

MAJOR:

• Abstract – the quality of the abstract does not reflect the quality of the paper. Please revise.

Our answer: We have revised our abstract as suggested by Reviewer #2.

• Methods – There are somethings that are presented in the results section that were not clarified in the methods, please ensure that all outcome metrics presented in the results are described in the methods.

Our answer: To acknowledgment reviewer’s remark, we have described all the variables needed in the methods section to justify our findings. 

• There are a many outcomes presented in this paper, which is great and demonstrates a very thorough investigation of the research question. However, at times, keeping track of all these outcomes is a bit confusing, especially with the many different tables. Throughout the methods and results, could be helpful to clearly map out the outcomes being measured and ensure that the results presented map to these – for example, where does satisfaction fit in?

Our answer: To keep track of all our results according to the method used, we have added and described three sub-points (practicability, accuracy, and preference) in the “Data collection tools” and two sub-points (observation of manipulation and interpretation of results) in the “field procedures” sections. We also have modified all concerns regarding the methods. Thus, we feel that our methods are more explicit in explaining our results. 

• Re-read the paper for grammar and language, sometimes the language used to describe things is a bit unusual and inconsistent. Try to use simple, consistent language when possible.

Our answer: The English of the manuscript has been fully edited by the assistance of International Research Promotion English Language Editing Services (IRP-ELES) to fully copyedit our manuscript in order to ensure proper English wording and grammar.

MINOR:

Abstract:

• (Background): Suggest “HIVST is an additionally approach…”

Our answer: As suggested, we have corrected the sentence. 

• (Background): Suggest taking out details on the test manufacturer in the abstract – can leave this for the paper.

Our answer: As suggested, we have corrected the sentence. 

• (Methods): What is your study population – what was the eligibility criteria for inclusion in the study?

Our answer: To answer the reviewer's question, we have added the following sentence to the method section of the Abstract: “Volunteers from the general public at high risk of HIV infection, between 18 and 49 years of age, and who had signed an informed consent form were eligible for the study.” 

• (Methods): Not clear if qualitative or quantitative data was collected – please clarify.

Our answer: Our study was quantitative using a structured questionnaire. We made these clarifications in the manuscript. 

• (Methods): You define practicability, but not accuracy and preferences (discussed in the background). Please clarify how these other outcomes were measured.

Our answer: We have clarified in the revised version all outcomes measured. 

• (Methods): Not sure what you mean by matching tests, please provide more information.

Our answer: To acknowledgment reviewer’s remark, we have corrected this sentence to avoid any ambiguity. 

• (Results): How were participants prospectively enrolled if this was a cross-sectional study?

Our answer: To acknowledgment reviewer’s remark, we have corrected this sentence to avoid any ambiguity. 

• (Methods/Results): How are you defining a successful performance? % of certain necessary steps completed?

Our answer: The successful performance of the HIV self-test was conditioned by the presence of the control band (valid results). We have added this clarification in methods section of the abstract. 

• (Results): What % of the tests resulted in invalid results?

Our answer: As the percentages of valid tests corresponded to the successful performance rate, the percentages of invalid tests were 0.4% (n=2) for the blood self-test and 0.6% (n=3) for the oral-fluid self-test. We have not added these results in the abstract because the corrected version has already clearly defined the variables. 

• (Results): How did you measure sensitivity and specificity?

Our answer: We have clarified the sensitivity and specificity measure in Abstract section as follow: “The field sensitivity and specificity of the results of the HIV self-test performed and interpreted by the participants compared to reference HIV testing results were estimated to evaluate the accuracy.”

• Check punctuation and spacing between words throughout.

Our answer: We have checked all punctuation and spacing between words. 

• (Conclusions): Consider mentioning how there was a greater preference for oral-fluid versus blood-based HIVST.

Our answer: We mentioned this result in the conclusion as follow: “Although preferences for oral fluid-based HIVSTD are high in this study, both approaches appear to be complementary, leaving users with preferences for each test for similar results.”

Methods:

• (Study design): How can a cross-sectional survey have a semi-structured questionnaire? These are usually used in qualitative research. Please explain.

Our answer: As explain above, our study was quantitative using a structured questionnaire. To acknowledgment reviewer’s remark, we have corrected “semi-structured questionnaire by “structured questionnaire”. 

• (Study setting): It does not seem that the locations of the study was “arbitrary” if you have a rationale for these settings, which you appear to have.

Our answer: We thank the reviewer for this remark. We have removed “arbitrary”. 

• (Study population): How did you determine that participants were at high risk of HIV infection? How did you recruit participants into the study?

Our answer: We have moved the definition of high risk back to this section as suggested below. We have also clarified in the manuscript how we recruited the participants.

• (Data collection): Same question as above about semi-structured survey. Suggest moving this information about “high risk” to the study population section above.

Our answer: We have moved the information concerning “high risk” to the study population section as suggested. 

• (Data collection): Is correctly interpreting results a practicability measurement or an accuracy measurement?

Our answer: The correct interpretation of the results was the measurement of both practicability and accuracy in this study. 

• (Data collection); What do you mean by reference HIV testing results? Can you be more specific here?

Our answer: The reference HIV testing results or the baseline HIV test results were the laboratory results using the national HIV testing algorithm. We have indicated this the manuscript. 

• (Field procedures): When you screened participants using the Determine test, did you give them their test results? Or did they just receive their test results from the self-testing kits? Additionally, do you have photos of the other standardized tests you had them interpret? (include as appendices?)

Our answer: Participants were told to consider only the results of HIV testing according to the national HIV screening algorithm because the HIV self-test should be used only for research purposes. Other standardized tests were the cassettes of Exacto Test HIV et OraQuick HIV Self-test that were previously performed in the lab with positive and negative samples for constituting the panel including two cassettes with positive results, two cassettes with negative results, and one invalid cassette (not analyzed). Unfortunately, we have not made picture of these standardized tests (Exacto and OraQuick).

• (Data management and analysis): I would move the information about the satisfaction criteria to the data collection section. Also, for your multivariable logistic regression model, what independent variables did you include in this analysis?

Our answer: To acknowledgment reviewer’s remark, we have moved the information about the satisfaction criteria to the data collection section. We have also indicated the independent variables includes in the multivariable logistic regression as following: “Finally, when identifying the independent predictors of the participants’ preferences for Exacto Test HIV and/or the OraQuick HIV Self-test, variables with a P-value < 0.2 in bivariate analysis and variables identified from the literature were entered into a full multivariate logistic regression model.”

Results:

• (Study participants): Do you mean no knowledge of the existence of HIVST?

Our answer: Yes, we have corrected in the manuscript.

• (Table 1): What is the number for the overall participants not the sum of the non-reactive and reactive sorting test participants?

Our answer: It was a mistake, we have corrected it. 

• (Practicability): How did you calculate the usability index? This should be in the methods section.

Our answer: The usability index was calculated as the arithmetic mean of all the correct answers from when the manipulation was observed. We have added this explanation in the statistical analysis paragraph of the Methods section. 

• (Table 2): Can the information presented in this table be simplified? Can you just include the “Yes” column for the steps and abbreviate the text describing the steps?

Our answer: We have corrected Table 2 as suggested.

• (Practicability): What were the outcomes of the tests? E.g., how many were inconclusive?

Our answer: When extracting the results of the tests performed and interpreted by the participants, 268 (50.7%), 258 (48.9%), and 2 (0.4%) participants had interpreted their results with the blood self-test as positive, negative, and invalid, respectively; and 268 (50.7%), 257 (48.7%), and 3 (0.6%) participants had respectively interpreted their results with the oral fluid self-test as positive, negative, and invalid, respectively. Misinterpretation concerned two positive results with a low-intensity test band with Exacto Test HIV (misinterpreted as negative result by participants) and five (3 weak positive and 2 negative misinterpreted as negative and positive result by participants, respectively) results with OraQuick HIV Self-test. All invalid results (n = 5) of performed HIV self-tests in this series were correctly interpreted. Thus, Cohen's � coefficients herein between the results of reading by participants versus observers were 0.99 and 0.98 for the blood test and oral-fluid test, respectively. We have added this results in the manuscript. 

• (Table 3): Can you add a description of how you calculated the Youden index to the methods section? 

Our answer: As suggested, we have added the calculation of the Youden index to the methods section. 

• (Accuracy): For this measurement, can you clarify if this is based on participant interpretation of the HIVST, or researcher interpretation of the HIVST?

Our answer: The sensitivity and specificity of the two self-tests were calculated based on the participants' interpretation of the self-test results. We have added this clarification in the manuscript. 

Discussion:

• What discussing the accuracy of the HIVST compared to standard testing methods, can you also discuss how self-interpreted results can decrease the accuracy of the HIVST, because participants might interpret the results differently from what the test is (accurately) showing.

Our answer: We have discussed this concern in the revised version of our manuscript.

---

## [Decision Letter · Decision Letter 1]

10 Sep 2020

Field evaluation of capillary blood and oral-fluid HIV self-tests in the Democratic Republic of the Congo

PONE-D-20-17754R1

Dear Dr. Tonen-Wolyec,

We’re pleased to inform you that your manuscript has been judged scientifically suitable for publication and will be formally accepted for publication once it meets all outstanding technical requirements.

Kind regards,

Joel Msafiri Francis, MD, MS, PhD

Academic Editor

PLOS ONE

Additional Editor Comments (optional):

Thank you for the revisions. Please address the minor additional suggestions from the reviewer.

Reviewers' comments:

Reviewer's Responses to Questions

**Comments to the Author**

1. If the authors have adequately addressed your comments raised in a previous round of review and you feel that this manuscript is now acceptable for publication, you may indicate that here to bypass the “Comments to the Author” section, enter your conflict of interest statement in the “Confidential to Editor” section, and submit your "Accept" recommendation.

Reviewer #1: All comments have been addressed

Reviewer #2: All comments have been addressed

2. Is the manuscript technically sound, and do the data support the conclusions?

Reviewer #1: Yes

Reviewer #2: Yes

3. Has the statistical analysis been performed appropriately and rigorously? 

Reviewer #1: Yes

Reviewer #2: Yes

4. Have the authors made all data underlying the findings in their manuscript fully available?

Reviewer #1: Yes

Reviewer #2: Yes

5. Is the manuscript presented in an intelligible fashion and written in standard English?

Reviewer #1: Yes

Reviewer #2: Yes

6. Review Comments to the Author

Reviewer #1: (No Response)

Reviewer #2: OVERALL: This is an important paper that will directly information HIVST implementation in DRC. The authors have been very responsive to reviewer comments and enhanced the quality of the paper. A few minor comments below.

MINOR:

• (Abstract): While the abstract has been revised to include a lot of great details, it is now quite long. Consider finding ways to consent the findings presented.

• (Methods, field procedures): Were participants give the results of the Determine test right away, or only after they completed the HIV self-test?

• (Results): Why were only 528 of 9776 (-127) participants included in the study?

• (Discussion, line 510): Do not think it is necessary to have a bolded “Strengths and limitations” at the beginning of this paragraph.

7. PLOS authors have the option to publish the peer review history of their article (what does this mean?). If published, this will include your full peer review and any attached files.

Reviewer #1: No

Reviewer #2: No

---

## [Editor Report · Acceptance letter]

25 Sep 2020

PONE-D-20-17754R1 

Field evaluation of capillary blood and oral-fluid HIV self-tests in the Democratic Republic of the Congo 

Dear Dr. Tonen-Wolyec:

I'm pleased to inform you that your manuscript has been deemed suitable for publication in PLOS ONE. Congratulations! Your manuscript is now with our production department. 

Kind regards, 

on behalf of

Dr. Joel Msafiri Francis 

Academic Editor

PLOS ONE